# A biome-wide experiment to assess the effects of propagule size and treatment on the survival of *Portulacaria afra* (spekboom) truncheons planted to restore degraded subtropical thicket of South Africa

Marius L. van der Vyver[1]☯*, Anthony J. Mills[2], Richard M. Cowling[1]☯

**1** Botany Department, Nelson Mandela University, Port Elizabeth, Eastern Cape, South Africa, **2** Department of Soil Science, Stellenbosch University, Matieland, Western Cape, South Africa

☯ These authors contributed equally to this work.
* mariusvdv@gmail.com

**Data Availability Statement:** All relevant data are within the manuscript and its Supporting information files.

## Abstract

Insights from biome-wide experiments can improve efficacy of landscape-scale ecological restoration projects. Such insights enable implementers to set temporal and geographical benchmarks and to identify key drivers of success during the often decades-long restoration trajectory. Here we report on a biome-wide experiment aimed at informing the ecological restoration of thousands of hectares of degraded subtropical thicket dominated by the succulent shrub, *Portulacaria afra* (spekboom). Restoration using spekboom truncheons has the potential to sequester, for a semi-arid region, large amounts of ecosystem carbon, while regenerating a host of associated ecosystem services. This study evaluates, after about three years post-propagation, the effects of spekboom truncheon size and treatment on survivorship in 40 fence-enclosed (0.25 ha) plots located in target habitat across the entire spekboom thicket biome. In each plot, locally harvested spekboom truncheons, comprising eight size/treatment combinations, were planted in replicated rows of between 24 and 49 individuals, depending on treatment. The experiment assessed the role of truncheon size, spacing, application of rooting hormone and watering at planting on survivorship percentage as an indicator of restoration success. All eight combinations recorded extreme minimum survivorship values of zero, while the range of extreme maximum values was 70-100%. Larger truncheons (>22.5 mm diameter) had almost double the survivorship (ca. 45%) than smaller truncheons (< 15 mm) (ca. 25%). Planting large, untreated truncheons at 1 m intervals—as opposed to 2 m intervals recommended in the current restoration protocol—resulted in no significant change in survivorship. The application of rooting hormone and water at planting had no significant effect on restoration success for both large and small truncheons. While our results do not provide an evidence base for changing the current spekboom planting protocol, we recommend research on the financial and economic costs and benefits of different propagation strategies in real-world contexts.

**Funding:** Marius van der Vyver and Richard Cowling are grateful funding from the Working for Woodlands Programme of the South African Department of Environmental Affairs and Tourism; National Research Foundation, Pretoria (grant number FA2005040700027 - Anthony Mills); and Nelson Mandela University. We acknowledge the invaluable contributions of the Subtropical Thicket Restoration Programme (STRP), the planting teams and managers (Gamtoos Irrigation Board) and the data sampling team (Conservation Support Services) as well as all landowners for allowing the experiment on their land. The funders had no role in study design, data collection and analysis, decision to publish, or preparation of the manuscript.

**Competing interests:** The authors have declared that no competing interests exist.

# Introduction

Many restoration projects implemented at the scale of landscapes seldom achieve their stated goals of restoring biodiversity and ecosystem services to pre-degradation levels [1–3]. Low restoration efficacy has been linked to a wide range of factors mostly acting in concert [4]. These include site factors such as variation in soil type or local climate, landscape context such as the proportion of landscape that is degraded, differences in degradation trajectories, and inadequate funding or political will to sustain interventions [5–9] Given the complexity of these factors, effective restoration practice typically requires planning at fine spatial scales [8, 10, 11], which combines expertise from a wide range of disciplines [12]. Also crucial is a commitment to routine monitoring and evaluation [13–15], which ideally should be expressed as a quantitative valuation of restoration outcomes [16] to evaluate efficiency and cost-benefits at certain stages along a commonly decades-long restoration trajectory [17–19]. In this regard, robust, biome-wide experiments—albeit costly and difficult to implement—may be critical for establishing benchmarks to evaluate efficiency and effectiveness over time and space, as well as to identify the main factors influencing the restoration process.

Payments for ecosystem services, including carbon sequestration, are potentially viable options to finance restoration action on a landscape-scale in many contexts [2, 20–24]. In some ecosystems, increasing carbon storage post restoration, both above- and belowground, is fundamental to achieving state transition and can be used as a proxy for restoration efficacy during the first decades of the restoration process [2, 17, 25–28]. Such opportunities are promising despite the uncertainty surrounding the global carbon price and the implementation of carbon taxation [29, 30]. In contexts where restoration outcomes have strong socio-economic imperatives, it is useful to measure effective restoration in terms of monetary costs and benefits [4, 31, 32]. Thus, cost-effectiveness, quantification of benefits and losses and related timescales are of prime concern [3, 33–36]. This is true in the context of this study, a developing country with high rates of poverty, where restoration initiatives must "earn their keep" [37].

Subtropical thicket dominated by *Portulacaria afra* (hereafter referred to as spekboom thicket)—a leaf succulent shrub to low tree—forms part of South Africa's Subtropical Thicket biome [38, 39]. Characterized as a dense, tangled mass of low trees (up to 5 m in height) and large shrubs—which are often spinescent, succulent, or both— spekboom thicket has been extensively degraded by injudicious land-use practices, particularly through over-browsing by domestic goats [40–43] (Fig 1). Approximately 45% of spekboom thicket (5,519 km$^2$ out of a total of 12,624 km$^2$) had been altered in this manner by 2000 [44].

Spekboom can shift between C3 photosynthetic and Crassulacean Acid Metabolism (CAM) in response to increased water stress, longer photo periods and increased daytime temperatures [45–47]—thereby retaining productivity during drought periods. It is highly palatable and widely used as a fodder plant by mega-herbivores such as elephant (*Loxodonta africana*), buffalo (*Syncerus caffer*), black rhinoceros (*Diceros bicornis*) and greater kudu (*Tragelaphus strepciseros*), as well as domestic stock such as goats and cattle. It is regarded as the backbone of southeastern South Africa's highly lucrative angora mohair industry [48] and forms the foundation of the regions thriving wildlife and ecotourism economies [49].

Intact spekboom thicket stores carbon of about 200 t·ha$^{-1}$ (measured up to a soil depth of 50 mm)—a remarkable feature for a xeric ecosystem [50, 51]. Most of this carbon stock is associated with spekboom litter fall [52, 53] its dense canopy provides the relatively cool and dry conditions necessary for the accumulation of the high levels of soil carbon [54–56] and maintenance of biodiversity [49, 57]. Comparisons of degraded and intact sites reveal carbon losses of more than 80 t C·ha$^{-1}$ [50, 58]. These losses are evident from the decrease in aboveground biomass Fig 1, but also manifest as a massive reduction in soil organic carbon content [59, 60].

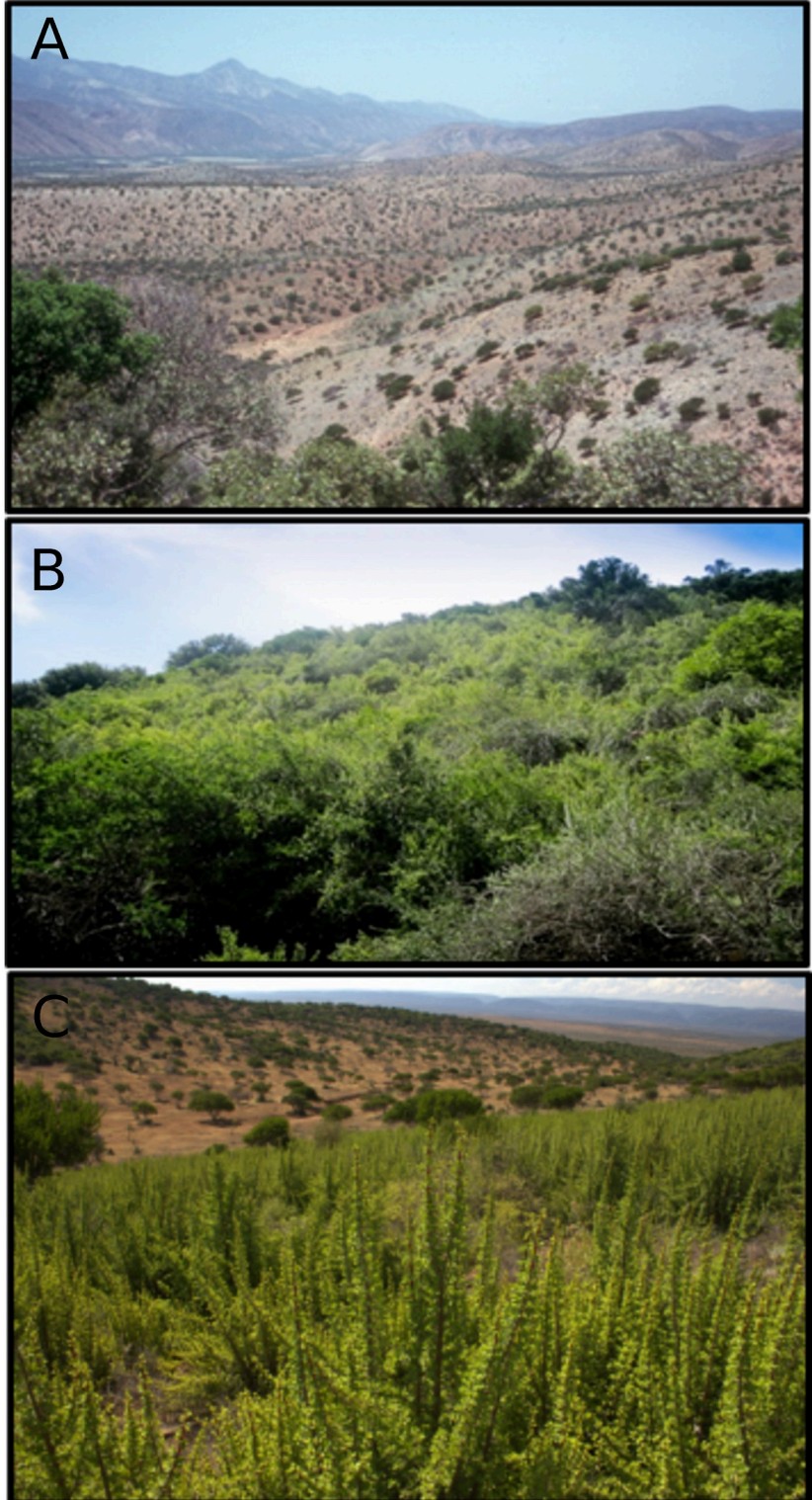

**Fig 1. Degraded and intact spekboom sites compared with a 3-year old restoration site.** A: a foothill landscape comprising livestock-degraded spekboom (*Portulacaria afra*) thicket where all spekboom has disappeared, leaving only woody canopy shrubs of *Pappea capensis* and *Euclea undulata* with a field layer of largely ephemeral plants. B: intact spekboom thicket (2-3 m tall) located in the same region as the top photo. Note the dense canopy of spekboom, scattered *P. capensis* trees on the skyline, and absence of bare ground (photos: RM Cowling). C: a thicket-wide plot

(TWP) located in target habitat and subject to low browsing intensity. Note the good establishment and growth of spekboom truncheons approximately three years after planting (photo: ML van der Vyver).

Under current environmental conditions, degraded spekboom thicket does not regenerate spontaneously to its intact state [40, 41, 54, 55]. However, planting spekboom truncheons does restore populations of this species [17, 52], sequester relatively large amounts of carbon [17, 52, 61] and—after 30–50 years—facilitate the restoration of other members of intact spekboom thicket, including long-lived trees and shrubs [17]. Indeed, after this period of restoration, the restored state resembles the intact state in terms of biodiversity, vegetation structure and ecosystem services. However, the transition from degraded to intact states requires the replenishment of above- and belowground carbon, which is achieved using the ecosystem engineering properties of spekboom [17]. Stands under restoration older than five years are scarce, and studies quantifying appropriate reference sites in terms of biodiversity, ecosystem services, structure and function are few. Research on established (5-50 year) stands under restoration report annual total carbon sequestration rates of between 1.4 and 4.5 t C ha$^{-1}$ yr$^{-1}$ [17, 52, 62]. These data were derived from restoration stands which were informally established by landowners and hence, lack information on implementation techniques such as truncheon size, planting depth, planting season and truncheon spacing.

To inform best-practice for the biome-wide restoration of spekboom thicket, the Subtropical Thicket Restoration Project (STRP)—a community of scientists, government officials and implementers [50]—established in 2007 a thicket-wide plot (TWP) experiment, comprising 300 plots (Fig 1), over the entire range of spekboom thicket. A major goal of the TWP experiment was to evaluate different truncheon planting techniques to identify the most efficient and effective protocols for maximising annual carbon sequestration rates throughout the restoration trajectory. Using 173 of the thicket-wide plots described above, Van der Vyver et al. [63] applied a rule-based learning ensemble to identify the physiographic and management-related factors that best predict restoration efficacy. Browsing impacts and the misidentification of target habitat (i.e. location of plot in non-spekboom thicket vegetation) emerged as the two most important predictors, negatively affecting both survivorship and carbon sequestration. However, this study was restricted to only one of the treatments applied to the spekboom truncheons—representative of the current restoration protocol used by the STRP—namely planting large (22.5 mm diameter) truncheons every two meters with no applications of root hormone or water at planting (Treatment 2 in Table 1).

Understanding the effects of different truncheon planting methods is of great importance to the STRP. For example, while planting larger truncheons may positively influence restoration efficacy, efficiency may be compromised relative to planting smaller truncheons which, because of their abundance in source sites and requirement for shallower holes, are likely less costly to harvest and to plant. Similarly, while watering at planting may increase survivorship, the effort and costs of getting water to remote restoration sites may well outweigh this benefit. In this paper, we focus on the effectiveness in terms of truncheon survivorship, or the benefit side, of different planting methods, and refer to costs only indirectly and qualitatively. Here, we assess combinations of eight treatment/size combinations (hereafter treatments) to explore the role of truncheon size (measured as stem diameter), spacing of truncheons, application of rooting hormone and application of water at planting on truncheon survivorship, approximately 3.5 years after planting. For this analysis, we selected the subset of plots (n = 40) that were in target habitat (spekboom thicket) and exposed to low browsing intensity—factors that promote restoration success of spekboom [63]. While the post-restoration period is sufficiently long to accommodate the post-planting mortality peak, it is too short to provide a useful

**Table 1. Treatments applied to truncheons of spekboom (*Portulacaria afra*) planted in 40 plots (0.25 ha) located across the extent of spekboom thicket vegetation (see Fig 2).**

| Treatment | Mean no. (range) of alive truncheons per plot | Stem diameter (mm) | Planting depth (mm) | Additional treatment |
|---|---|---|---|---|
| 1 | 85.8 (165-5) | 30 | 30 | None |
| 2 | 78.8 (145-5) | 22.5 | 30 | None |
| 3 | 30.3 (70-1) | 22.5 | 30 | 2-m spacing |
| 4 | 75.2 (132-5) | 22.5 | 30 | Rooting hormone |
| 5 | 78.3 (150-11) | 22.5 | 30 | Water at planting |
| 6 | 22.9 (56-0) | 15 | 15 | None |
| 7 | 26.6 (67-3) | 10 | 10 | None |
| 8 | 29.3 (87-0) | 10 | 10 | Water at planting |

**Notes**: Other than Treatment 3, all truncheons were planted at 1 m intervals.

assessment of patterns of carbon sequestration in relation to truncheon treatment; this will be undertaken shortly, now that the experiment has run for more than a decade.

## Study area

The study area comprises the global distribution of spekboom thicket, delimited by the Fish River drainage in the east and the Gouritz River drainage in the west (Fig 2) [39]. This region is highly heterogeneous in terms of topography, climate, geology and soils and is home to eight biomes which interdigitate in complex ways [39, 64]. Spekboom thicket is often, but not exclusively, associated with relatively deep, shale-derived and mostly rocky soils on sloping ground [39]. Annual rainfall is bimodally distributed, with peaks in spring and autumn. In the west, more rain falls in the cooler months while the east may experience appreciable summer

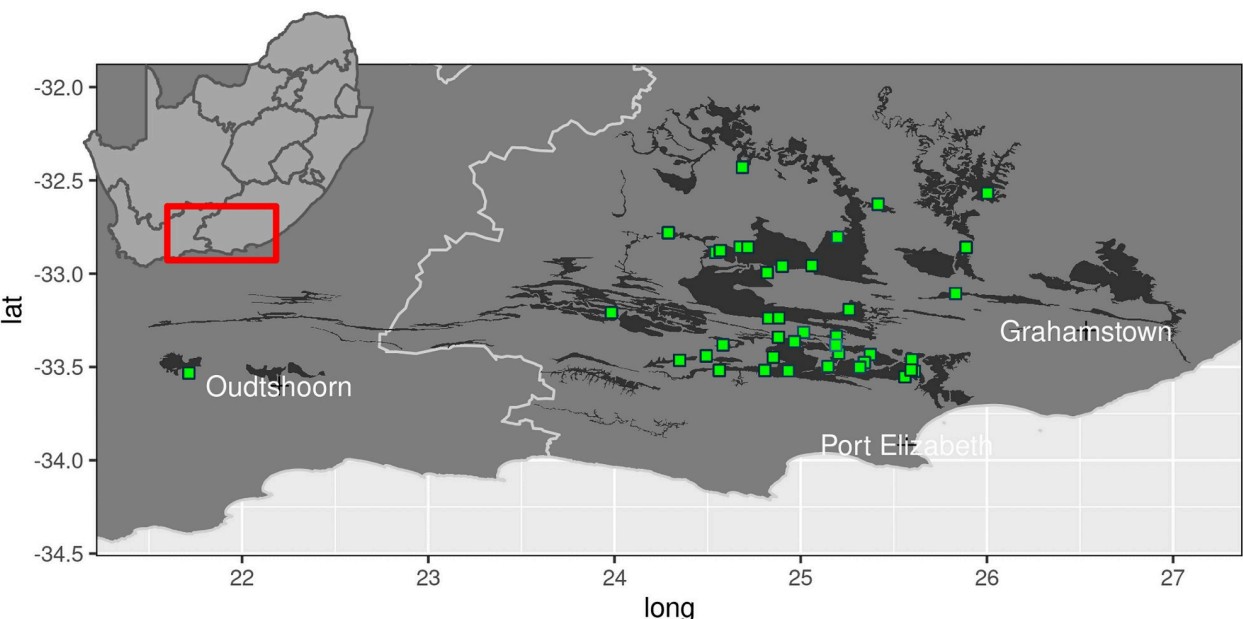

**Fig 2. Map of the study region showing the extent of the spekboom thicket vegetation (dark shading) and location of the 40 experimental plots used in this study (green squares).**

rain. Mean annual precipitation ranges from 250 mm in the western, inland regions to 550 mm in the eastern, coastal regions [48]. Localised flash-floods and prolonged droughts are not uncommon. Summers are very hot with temperatures often exceeding 40˚C; winters are mild, but the nights may be cold and frost occurs regularly in bottomlands because of cold-air pooling [65–69]. Spekboom thicket eschews frost-exposed terrain [67, 68]. The geology is dominated by rocks associated with the Cape Supergroup (quartzitic sandstones, shales), Karoo Supergroup (mudstones, shales) and Uitenhage Group (mudstones, conglomerates). Other than Cape sandstones, which yield infertile sands, other geologies yield mainly moderately fertile and relatively clay-rich soils [64, 70, 71].

## Materials and methods

### Experimental plots

Our study used 40 of 300 0.25 ha plots established throughout the spekboom thicket biome between March 2008 and October 2009 by unskilled teams contracted and trained by the national Department of Environmental Affairs' Working for Woodlands programme via its implementing agent, the Gamtoos Irrigation Board (for more details, see van der Vyver et al. subm.). Managers and contracted teams were given training to enable them to harvest spekboom cuttings of suitable dimensions from nearby intact stands and plant these in multiple rows, according to different treatments within a fenced herbivore exclosure of 0.25 ha (50 x 50 m) (Fig 1). Fencing comprised a 1.2 m high wire-mesh stabilised by steel droppers and wooden posts at the corners. Two strands of wire were added to increase the height to circa 1.4 m to prevent access by domestic stock and potentially deter (but not eliminate) entry by the large, indigenous browser, the greater kudu (*Tragelaphus strepsiceros*).

In each plot, the eight treatments were replicated by 2–4 rows, comprising 49 truncheons (27 truncheons per row in Treatment 3 where truncheons were planted at two-meter intervals) (Table 1). Treatment 3, comprising truncheons of 22.5 mm diameter, planted at 30 mm depth with no additional treatment, is the current planting protocol for landscape-scale restoration projects implemented by the STRP. Other treatments were distinguished according to stem diameter of truncheons and the application of extraneous treatments, namely the root hormone 4-Indole-3-Butyric Acid (IBA) and water (200 ml) at planting. We did not vary planting depth systematically. Given the widespread stoniness of spekboom soils, the excavation of planting holes is time-consuming and expensive. Thus, we applied planting depths commensurate with maintaining the truncheon in an upright position in the face of wind, rain and browsing. These depths are 30 mm for larger truncheons and 10–15 mm for smaller ones (Table 1).

Treatment rows were assigned randomly. This design created problems for data collection as treatments in many plots could not be identified, owing to poor database management. Thus, only 162 plots were retrieved where all treatments could be identified and sampled. Of these, we focused here on the 40 plots that were in the correct target habitat (spekboom thicket and associated ecotones that were identified in the field by an expert thicket ecologist) and subject to a moderate or low browsing intensity. This was done because these were the explanatory variables identified in rule-based, predictive models as having the strongest positive effects on both survivorship and carbon sequestration of one of the treatments (Treatment 3 in Table 1) [63].

### Data collection

The response variable for this experiment was survivorship percentage of planted truncheons. Data were collected by a scientific services company (Conservation Support Services) between

**Table 2. Combinations of treatments for assessing impacts of a specific treatment on the survivorship of spekboom (*Portulacaria afra*) truncheons (see Table 1 for description of treatments).**

| Target treatment | Description | Treatment combinations1 | Planting depth (mm) |
|---|---|---|---|
| Stem diameter | 30, 22.5, 15 & 10 mm diameter | 1,2,6,7 | 1 & 2 = 30 6 & 7 = 15 |
| Spacing | 1 or 2 m 22.5 mm diameter | 2,3 | 30 |
| Rooting hormone | Yes or no 22.5 mm diameter | 2,4 | 30 |
| Water at planting | Yes or no 22.5 & 10 mm diameter | 2,5 & 7,8 | 30 & 10 |

June 2012 and January 2013, some 33–57 months (3–5 years) after planting, depending on when a particular plot was planted. Thus, all plots were at least about three years post-planting at the time the survivorship data were collected, by which time mortality rates would have stabilised. Survivorship of planted truncheons per treatment was quantified by counting all alive, dead and missing truncheons in all planted rows in each plot.

## Experimental design and statistical analysis

We combined different treatments (Table 2) to test hypotheses on the effects on survivorship of stem diameter, spacing of truncheons, application of rooting hormone and addition of water at planting. Survivorship was predicted to be positively related to stem diameter, since greater reserves in the succulent stems of larger truncheons would confer greater resistance to bouts of prolonged water stress post-planting [72, 73]. We also expected that the denser 1 m spacing would—owing to intraspecific competitive effects typical of many semi-arid communities [74, 75]—reduce survivorship, thereby reducing the benefits for carbon sequestration of their higher density. Finally, we posited that the application of rooting hormone and water at planting would improve survivorship [63, 76], specifically for smaller truncheons with fewer reserves. Survivorship data were normally distributed for all analyses. Hence, we used parametric statistics to test hypotheses (one-way ANOVA and Welch two sample t-tests). We used base R [77] to implement all statistical analyses, while graphics were created with the tidyverse [78] and ggplot2 [79] packages.

## Results

We recorded significant variation in survivorship across all treatments for (F = 12.97; P< 0.0001) (S1 Fig). All treatments recorded extreme minimum survivorship values of < 5%, while extreme maximum values of > 90% survival were observed in three treatments, including one using small truncheons. Treatment 1 (30 mm diameter truncheons with no additional treatment) (Table 1) recorded highest mean survivorship (44%) whereas lowest mean survivorship (24%) was recorded for Treatment 6 (15 mm diameter truncheons with no additional treatment).

Survivorship was significantly influenced by truncheon size (F = 11.96; P< 0.0001). As we predicted, larger truncheons (> 22.5 mm diameter) had significantly higher (almost two-fold) survivorship (ca. 45%) than smaller truncheons (< 15 mm diameter) (ca 25%) (Fig 3). However, differences between the two larger diameters (Treatments 1 and 2, see Table 1) were not significant, nor were differences between the two smaller diameters (Treatments 6 and 7). This implies that a truncheon diameter of 22.5 mm is a likely low-size threshold for maximizing survivorship of spekboom truncheons in restoration projects. When compared to the STRP restoration protocol of planting truncheons of 22.5 mm diameter at 2 m intervals (Treatment 3 in Table 1), decreasing the planting interval to 1 m (Treatment 2) had no effect (t = -0.017; P = 0.99) on survivorship (Fig 4).

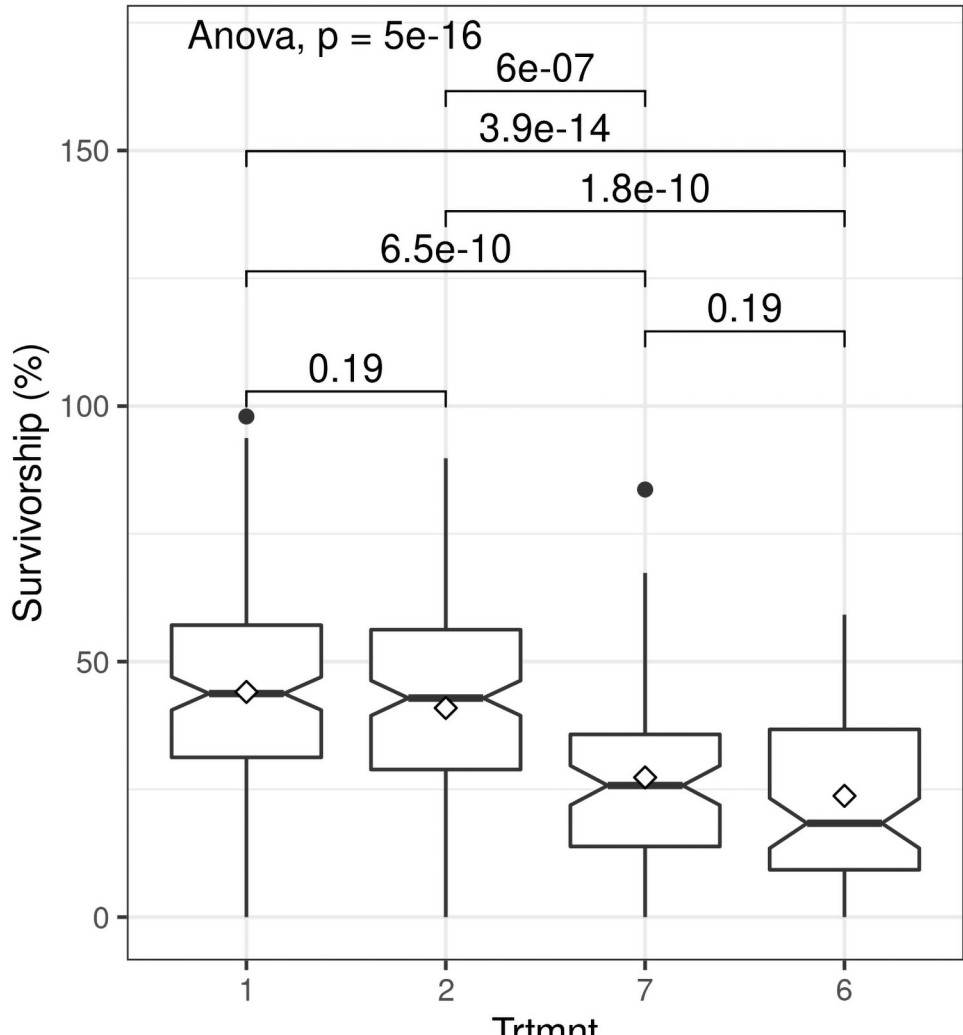

**Fig 3. Boxplots showing the effect of stem diameter of planted spekboom (*Portulacaria afra*) truncheons on percentage survivorship for different stem diameters (Trtmnt) (1 = 30 mm, 2 = 22.5 mm, 6 = 15 mm and 7 = 10 mm; see Tables 1 and 2 for details).** Within-box horizontal lines depict median values and the diamonds depict mean values. The paired comparisons above the boxes are P values for t-tests.

Applying rooting hormones to truncheons of 22.5 mm diameter had no significant effect on survivorship (t = 0.99; P = 0.32) (Fig 5) nor did the application of 200 ml of water prior to planting of truncheons with diameters of 10 mm (t = -1.19; P = 0.24) and 22.5 mm (t = 0.26; P = 0.079) (Fig 6).

## Discussion

Our study shows that over the duration of the experiment, survivorship was maximised by planting large (22.5 and 30 mm diameter) truncheons, irrespective of the application of additional treatments. Treatments using larger truncheons recorded a survivorship of ca. 45%, almost twice as high as those using smaller truncheons. Extreme upper values of > 90% survivorship that we recorded for three treatments (two large-truncheon and one small-truncheon treatments) represent a high level of success for perennial plant restoration in semi-arid

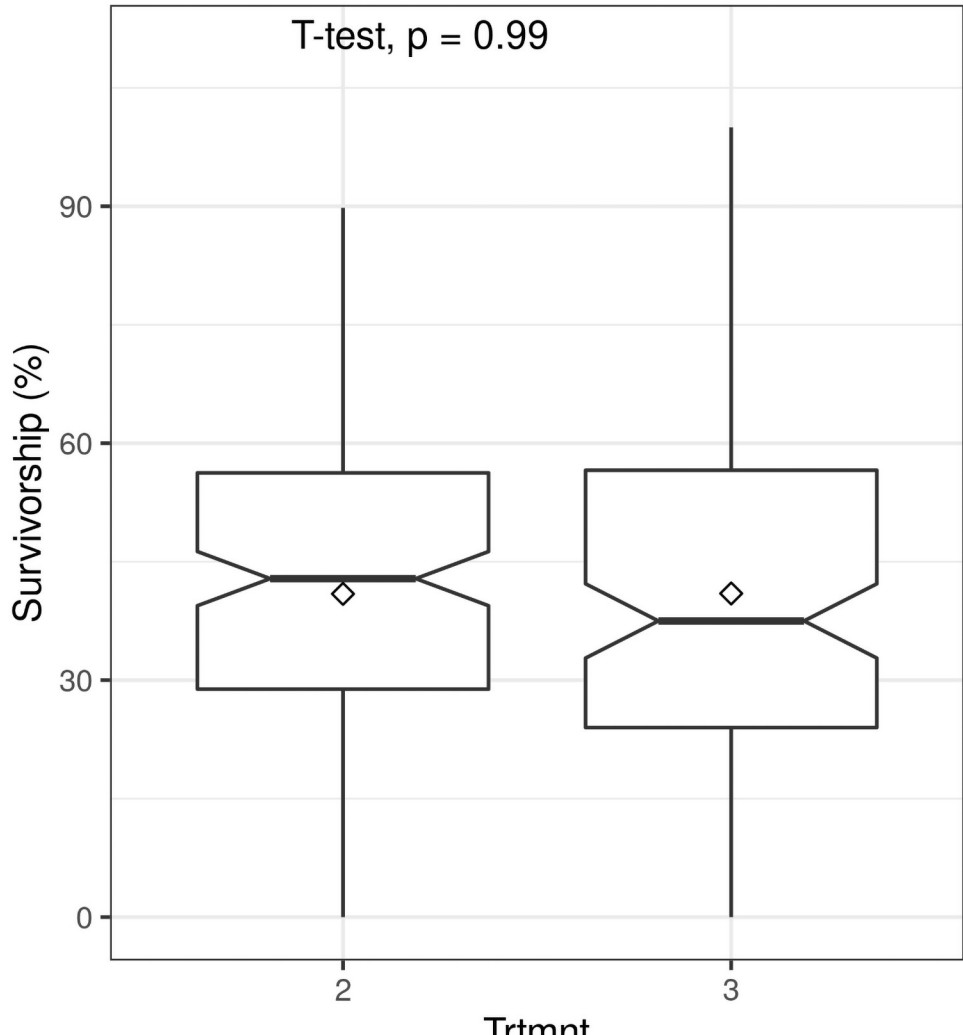

**Fig 4. Boxplots showing the effect of spacing at planting of spekboom (*Portulacaria afra*) truncheons of 22.5 mm diameter on survivorship (Trtmnt 2 = 1-m spacing, Trtmnt 3 = 2-m spacing; see Tables 1 and 2 for details).** Within-box horizontal lines depict median values and the diamonds depict mean values.

environments, where poor performance is the norm [80–84]. In their review of 54 restoration experiments across a range of ecosystems, [83] showed that nursery-grown seedlings used in restoration projects had a mean survivorship of 62% whereas survivorship of seedlings emerging from in situ sowing of seeds was only 18%. In a semi-arid setting comparable to the context of this study, survivorship of the best performing 3-month old seedlings of three shrubs barely reached 31%, even after added moisture [80]. Indeed, planting cuttings instead of transplanting containerised rooted seedlings grown in a nursery, is generally less successful in terms of survivorship [85]. When carrying out such planting activities, it is important to note that the defining characteristic for survivorship of planted propagules is root to shoot ratio. Sufficiently developed root systems of transplanted seedlings or even bareroot seedlings confer better drought tolerance [86, 87] than stem cuttings with no roots. In this respect, the survivorship values we recorded in this study for large truncheons are remarkably high, especially when

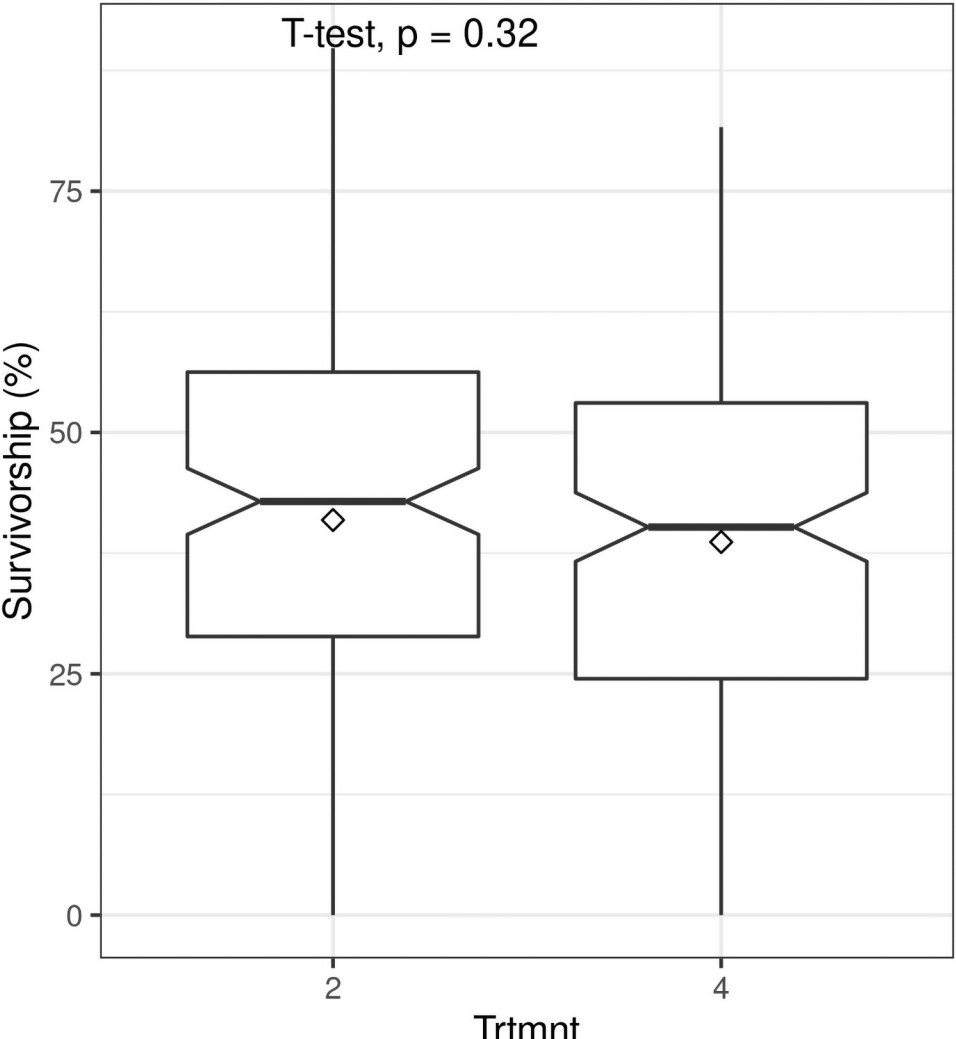

**Fig 5. Boxplots showing the effect on survivorship of root hormone application at planting (Trtmnt 2 = no hormone, Trtmnt 4 = hormone application; see Tables 1 and 2 for details) of spekboom (*Portulacaria afra*) truncheons of 22.5 mm diameter planted at 1 m intervals.** Within-box horizontal lines depict median values and the diamonds depict mean values.

taking into consideration the reduced implementation costs of planting truncheons directly into the restoration site [17].

## Treatment effects

It is not surprising that, in a stem-succulent plant such as spekboom, larger truncheons performed better in restoration experiments than smaller truncheons. The larger the truncheon, the greater the amount of internally stored resources to weather droughts and tolerate exposure to inclement weather and browsing— as has been shown in other revegetation contexts [73, 88–90]. However, restoration using smaller truncheons is likely less costly in terms of truncheon harvesting, transport and planting, than using larger truncheons. Using smaller truncheons will also reduce the impacts of harvesting on intact sites used as a source of truncheons. Research is urgently required on the economic and ecological costs and benefits of

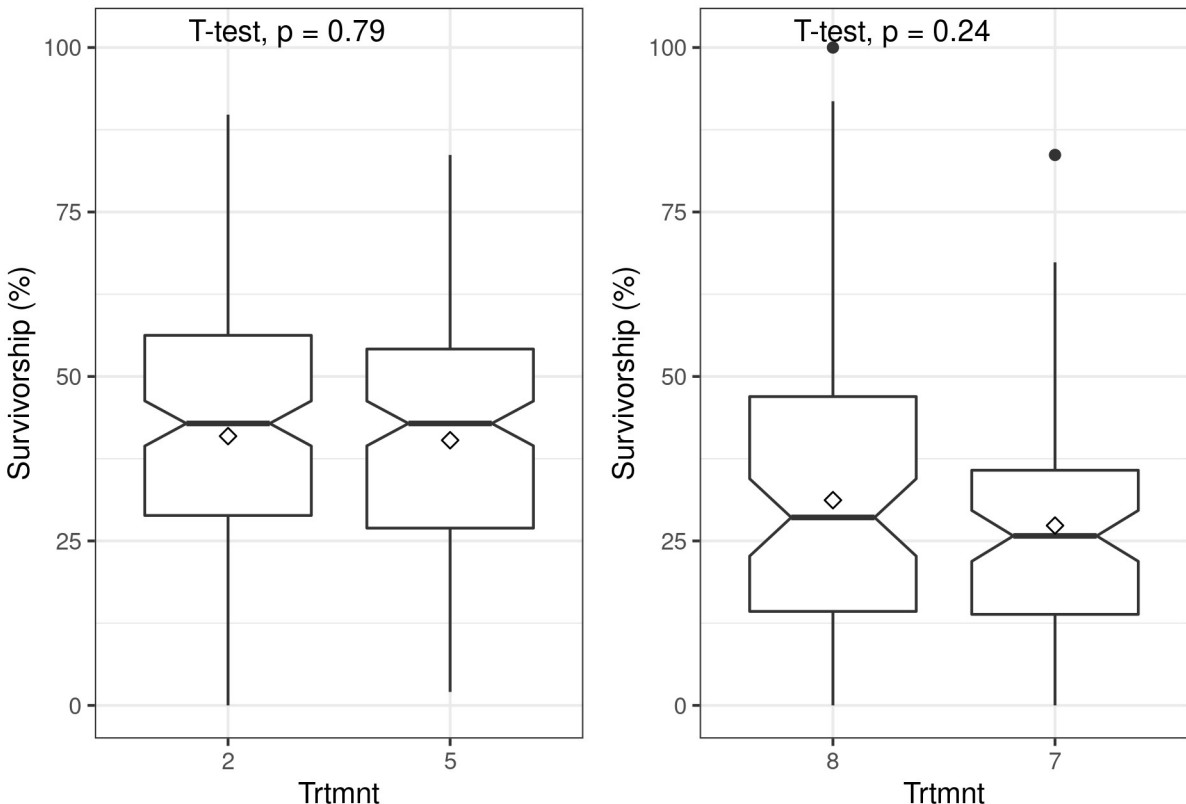

**Fig 6. Boxplots showing the effect of water application at planting of spekboom (*Portulacaria afra*) truncheons on survivorship.** The top panel is for 22.5 mm diameter truncheons and bottom panel is for 10 mm diameter truncheons, all planted at 1 m intervals. Treatments (Trtmnt) 5 and 8 had water application whereas treatments 2 and 7 had none (see Tables 1 and 2 for details). Within-box horizontal lines depict median values and the diamonds depict mean values.

planting different sized truncheons. We do not know the ecophysiological details of this truncheon size effect, another topic requiring further research. The almost two-fold higher survivorship of 22.5 mm versus 15 mm diameter truncheons is intriguing. The similar survivorship of truncheons of 30 mm and 22.5 mm diameter is also of interest. The practical implication of this is that the STRP protocol for selecting truncheon size can remain unchanged until rigorous cost-benefit analyses of using different truncheon sizes have been undertaken.

Doubling the planting density of large (22.5 mm diameter) truncheons relative to the STRP protocol showed no negative impacts on survivorship. This suggests that the likely increased (below-ground) competition among more densely planted truncheons does not affect survivorship. Planting truncheons less densely will also reduce the impact on intact sites used as a harvesting source of truncheons for restoration. However, planting at wider intervals may increase the time taken for spekboom canopy closure and the positive feedbacks this has for improving soil organic matter, reducing runoff and evaporation, enhancing soil water holding capacity facilitating biodiversity return [17, 55, 91]. Based on these data, we see no reason to change the STRP planting guidelines of 2 m spacing for 22.5 mm diameter truncheons. Although the experiment did not assess the potential impacts of planting density of smaller truncheons, we suspect that the lower carbon sequestration of planting these at 2 m intervals would outweigh the economic and ecological benefits of using smaller truncheons.

We found no positive effects of applying either rooting hormone or water-at-planting on survivorship, irrespective of truncheon diameter. This is surprising given the results from

other studies that recorded positive effects on restoration efficacy of applying these treatments [17, 92–94]. We would have expected, at least for the smaller (10 mm diameter) truncheons, that water-at-planting would have positively influenced survivorship [94]. Since rooting of cuttings of 17 succulent species takes between 21 and 53 days, depending on temperature [95], improved survivorship could be achieved by applying additional moisture three to four weeks after planting by which time most spekboom truncheons may have rooted. However, van der Vyver et al. [63] found only a weak, positive effect of the amount of rain falling post-planting on the survivorship of truncheons planted according to the STRP protocol.

## An improved protocol for restoring spekboom thicket

The array of treatments used in this study falls short of what is required to comprehensively inform an improved protocol for spekboom restoration. More research is required on extraneous treatments for maximizing survivorship, and spacing patterns for maximizing rates of canopy closure, with the concomitant increases in ecosystem service delivery, including the restoration of pre-degradation biodiversity [17]. Nonetheless, overall the STRP protocol of planting untreated, 22.5 mm diameter truncheons at 2 m intervals produced survivorship rates equivalent to the best among the other treatments. We see no reason to change this protocol until rigorous cost-benefit analyses have been undertaken for other treatments. Given the likely reduced costs, planting smaller truncheons shows great promise; we urgently recommend cost-benefit research using small truncheons for spekboom restoration.

## Supporting information

**S1 Fig. Box plots showing results for survivorship for all treatments (see Tables 1 and 2 for details).** Within-box horizontal lines depict median values and the diamonds depict mean values. (TIF)

**S1 Data.**
(CSV)

## Acknowledgments

We acknowledge the invaluable contributions of the Subtropical Thicket Restoration Programme (STRP), the planting teams and managers (Gamtoos Irrigation Board) and the data sampling team (Conservation Support Services) as well as all landowners for allowing the experiment on their land.

## Author Contributions

**Conceptualization:** Marius L. van der Vyver, Anthony J. Mills, Richard M. Cowling.

**Data curation:** Marius L. van der Vyver.

**Formal analysis:** Marius L. van der Vyver.

**Funding acquisition:** Marius L. van der Vyver, Richard M. Cowling.

**Investigation:** Marius L. van der Vyver, Anthony J. Mills, Richard M. Cowling.

**Methodology:** Marius L. van der Vyver, Anthony J. Mills, Richard M. Cowling.

**Project administration:** Marius L. van der Vyver, Richard M. Cowling.

**Supervision:** Anthony J. Mills, Richard M. Cowling.

**Validation:** Marius L. van der Vyver, Anthony J. Mills.

**Visualization:** Marius L. van der Vyver.

**Writing – original draft:** Marius L. van der Vyver.

**Writing – review & editing:** Richard M. Cowling.

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
