## [Decision Letter · Decision Letter 0]

18 Feb 2021

PONE-D-20-32304

A biome-wide experiment to assess the effects of propagule size and treatment on the survival of *Portulacaria afra* (spekboom) truncheons planted to restore degraded subtropical thicket of South Africa.

PLOS ONE

Dear Dr. van der Vyver,

Thank you for submitting your manuscript to PLOS ONE. After careful consideration, we feel that it has merit but does not fully meet PLOS ONE’s publication criteria as it currently stands. Therefore, we invite you to submit a revised version of the manuscript that addresses the points raised during the review process.

The reviewer has suggested for the minor revision. Please revise it accordingly.

We look forward to receiving your revised manuscript.

Kind regards,

Arun Jyoti Nath

Academic Editor

PLOS ONE

2. We note that Figure 2 in your submission contains map images which may be copyrighted. All PLOS content is published under the Creative Commons Attribution License (CC BY 4.0), which means that the manuscript, images, and Supporting Information files will be freely available online, and any third party is permitted to access, download, copy, distribute, and use these materials in any way, even commercially, with proper attribution. For these reasons, we cannot publish previously copyrighted maps or satellite images created using proprietary data, such as Google software (Google Maps, Street View, and Earth). For more information, see our copyright guidelines: http://journals.plos.org/plosone/s/licenses-and-copyright.

(1) You may seek permission from the original copyright holder of Figure 2 to publish the content specifically under the CC BY 4.0 license. 

Reviewers' comments:

Reviewer's Responses to Questions

**Comments to the Author**

1. Is the manuscript technically sound, and do the data support the conclusions?

Reviewer #1: Yes

2. Has the statistical analysis been performed appropriately and rigorously? 

Reviewer #1: Yes

3. Have the authors made all data underlying the findings in their manuscript fully available?

Reviewer #1: Yes

4. Is the manuscript presented in an intelligible fashion and written in standard English?

Reviewer #1: Yes

5. Review Comments to the Author

Reviewer #1: Comments on PONE-D-20-32304

Title: A biome-wide experiment to assess the effect of propagule size and treatment on the survival of Portulacaria afra (spekboom) truncheons planted to restore degraded subtropical thicket of South Africa by Marius et al.

The study reported a biome-wide experiment on Portulacaria afra to evaluate its survival and potential to restore the degraded subtropical thicket landscape of South Africa. The authors have done similar studies on the landscape with different objectives.

The manuscript is well written, with clear objectives and data interpretation and discussion are good. The language part of the manuscript is quite okay.

I suggest a few minor modifications/queries to be answered by the authors.

Materials and methods: L152, here plot size to be mentioned.

It is understood that the results are drawn from 40 plots, however, there is long description of 300 initial plots, subsequently 162 plots were retrieved and finally 40 plots were considered. The initial impression gave that the experiment sample size was very high. I feel the authors could mention that the results are based on experiment from 40 plots that had the correct target habitats. This study does not cover annual carbon sequestration of the target species (barring T3, that is the current planting protocol under STRP).

Clarity is required on L164-165. Do the authors mean that Treatment 3 used 27 truncheons and the remaining 7 treatments used only 22 truncheons out of the total 49 truncheons?

Data collection: L-189 mentioned data were collected after 33-57 months (3-5 years), while L-125 assessment on survival of truncheons were made after 3.5 years (approx.) after planting? The authors need to clarify if the plantation was done on the same time at different plots or at different time so as to make an assessment uniformly after 3.5 years of planting for survival of truncheons? Time of planting may play an important factor too to the survival of spekboom truncheons.

L-209: Figure no is missing.

Fig. 3. Trtmnt should be expanded while used for the first time.

Results: I would rather prefer the authors to write experimental results on the effect of size, space and planting depth on the survival of spekboom truncheons, and followed by the results on treatment combination on the survival of spekboom truncheons.

The list of references is too long, probably a few references may be deleted from the introduction part-but it is up to the authors!

I recommend acceptance the paper with minor revision.

6. PLOS authors have the option to publish the peer review history of their article (what does this mean?). If published, this will include your full peer review and any attached files.

Reviewer #1: No

---

## [Author Response · Author response to Decision Letter 0]

23 Mar 2021

PONE-D-20-32304: A biome-wide experiment to assess the effects of propagule size and treatment on the survival of Portulacaria afra (spekboom) truncheons planted to restore degraded subtropical thicket of South Africa.

Authors rebuttals in italics

Response to reviewers

Editor

You may seek permission from the original copyright holder of Figure 2 to publish the content specifically under the CC BY 4.0 license. 

Figure 2. M.L. van der Vyver created Figure 2 from the included dataset and freely available data derived from the South African National Biodiversity Institute (SANBI) free data repositories (https://bgis.sanbi.org/) and the South African Government (https://egis.environment.gov.za/gis_data_downloads).

Reviewer 

The study reported a biome-wide experiment on Portulacaria afra to evaluate its survival and potential to restore the degraded subtropical thicket landscape of South Africa. The authors have done similar studies on the landscape with different objectives. The manuscript is well written, with clear objectives and data interpretation and discussion are good. The language part of the manuscript is quite okay.

Thank you for these encouraging comments.

Materials and methods: L152, here plot size to be mentioned.

Done although we do note that plot size is mentioned lower down in the same paragraph.

It is understood that the results are drawn from 40 plots, however, there is long description of 300 initial plots, subsequently 162 plots were retrieved and finally 40 plots were considered. The initial impression gave that the experiment sample size was very high. I feel the authors could mention that the results are based on experiment from 40 plots that had the correct target habitats.

We feel it would be misleading not to mention how we arrived at 40 plots for the analysis in this paper. However, we have changed the first sentence of the paragraph by stating that we used only 40 of the 300 plots initially established.

This study does not cover annual carbon sequestration of the target species (barring T3, that is the current planting protocol under STRP).

Carbon sequestration rates are dealt with in another paper (van der Vyver et al. PeerJ in press) that is cited in our paper.

Clarity is required on L164-165. Do the authors mean that Treatment 3 used 27 truncheons and the remaining 7 treatments used only 22 truncheons out of the total 49 truncheons?

We have clarified our text by changing the text thus: “In each plot, the eight treatments were replicated by 2-4 rows, each row comprising 49 truncheons at 1-m spacing. However, for Treatment 3, where truncheons were planted at 2-m intervals, rows comprised 27 individuals.

Data collection: L-189 mentioned data were collected after 33-57 months (3-5 years), while L-125 assessment on survival of truncheons were made after 3.5 years (approx.) after planting? The authors need to clarify if the plantation was done on the same time at different plots or at different time so as to make an assessment uniformly after 3.5 years of planting for survival of truncheons? Time of planting may play an important factor too to the survival of spekboom truncheons.

Thank you for this useful comment. We agree that our wording lack clarity. Thus, we have added the following text to the sentence ending on L 190: “depending on when a particular plot was planted. Thus, all plots were at least about three years post-planting at the time the survivorship data were collected, by which time mortality rates would have stabilised”.

L-209: Figure no is missing.

Our typo. This statement does not refer to any figure.

Fig. 3. Trtmnt should be expanded while used for the first time.

Done

Results: I would rather prefer the authors to write experimental results on the effect of size, space and planting depth on the survival of spekboom truncheons, and followed by the results on treatment combination on the survival of spekboom truncheons.

We prefer to retain the organization of material in the Results section. Paragraph one provides a global (i.e. treatment-wide) overview of the results. The remaining paragraphs dig deeper into the results. We feel that this is a logical way to present the material.

The list of references is too long, probably a few references may be deleted from the introduction part-but it is up to the authors!

We have reduced these in cases of redundancy.

---

## [Decision Letter · Decision Letter 1]

5 Apr 2021

A biome-wide experiment to assess the effects of propagule size and treatment on the survival of *Portulacaria afra* (spekboom) truncheons planted to restore degraded subtropical thicket of South Africa.

PONE-D-20-32304R1

Dear Dr. van der Vyver,

We’re pleased to inform you that your manuscript has been judged scientifically suitable for publication and will be formally accepted for publication once it meets all outstanding technical requirements.

Kind regards,

Arun Jyoti Nath

Academic Editor

PLOS ONE

Additional Editor Comments (optional):

Reviewers' comments:

Reviewer's Responses to Questions

**Comments to the Author**

1. If the authors have adequately addressed your comments raised in a previous round of review and you feel that this manuscript is now acceptable for publication, you may indicate that here to bypass the “Comments to the Author” section, enter your conflict of interest statement in the “Confidential to Editor” section, and submit your "Accept" recommendation.

Reviewer #1: All comments have been addressed

2. Is the manuscript technically sound, and do the data support the conclusions?

Reviewer #1: Yes

3. Has the statistical analysis been performed appropriately and rigorously? 

Reviewer #1: Yes

4. Have the authors made all data underlying the findings in their manuscript fully available?

Reviewer #1: Yes

5. Is the manuscript presented in an intelligible fashion and written in standard English?

Reviewer #1: Yes

6. Review Comments to the Author

Reviewer #1: The authors have addressed all the queries raised, so the manuscript may be accepted for publication

7. PLOS authors have the option to publish the peer review history of their article (what does this mean?). If published, this will include your full peer review and any attached files.

Reviewer #1: No

---

## [Editor Report · Acceptance letter]

13 Apr 2021

PONE-D-20-32304R1 

A biome-wide experiment to assess the effects of propagule size and treatment on the survival of *Portulacaria afra* (spekboom) truncheons planted to restore degraded subtropical thicket of South Africa. 

Dear Dr. van der Vyver:

I'm pleased to inform you that your manuscript has been deemed suitable for publication in PLOS ONE. Congratulations! Your manuscript is now with our production department. 

Kind regards, 

on behalf of

Dr. Arun Jyoti Nath 

Academic Editor

PLOS ONE